# Quantum Cutting in Ultraviolet B-Excited KY(CO_3_)_2_:Tb^3+^ Phosphors

**DOI:** 10.3390/ma15176160

**Published:** 2022-09-05

**Authors:** Dechuan Li, Guangping Zhu

**Affiliations:** 1School of Physics and Electronic Information, Huaibei Normal University, Huaibei 235000, China; 2Key Laboratory of Green and Precise Synthetic Chemistry and Applications, Ministry of Education, Huaibei 235000, China

**Keywords:** quantum cutting, Tb^3+^, ultraviolet B, phosphors

## Abstract

Highly efficient quantum cutting KY(CO_3_)_2_:Tb^3+^ phosphors excited by ultraviolet B (UVB) and ultraviolet C (UVC) were investigated. The structural and spectroscopic properties were characterized by XRD analysis and fluorescence spectrophotometry, respectively. The results showed that the monoclinic crystal structure of KY(CO_3_)_2_:Tb^3+^ remained in the Tb^3+^ doping range of 0~100%. In the excitation spectrum, two intense excitation peaks were observed in the ultraviolet range. Under the excitation of 283 nm, the maximum quantum efficiency of KY(CO_3_)_2_:0.7Tb^3+^ could reach 119%. However, the most efficient quantum cutting occurred at the ^5^K_8_ excited state in the cross-relaxation of ^5^K_8_ + ^7^F_6_^5^D_4_ + ^5^D_4_. The Tb^3+^ content could be selected arbitrarily in the KY(CO_3_)_2_ host without any concentration quenching. Optimal quantum cutting concentrations of Tb^3+^ in KY(CO_3_)_2_ were 0.7 and 0.3 for the excitation of UVB and UVC, respectively. UVB-excited phosphors are more popular with high transparency in products such as glass or resin. A quick response code was fabricated by resin to show the hidden information clearly. Therefore, the highly efficient phosphor could be a candidate material for the application in information identification technology.

## 1. Introduction

Quantum cutting is the process of converting a single photon into two or more photons [1,2,3,4]. The quantum cutting efficiency is often larger than 100%. Vacuum ultraviolet and short-wave ultraviolet are often used as excitation sources for quantum cutting [5]. High-energy photons could excite ground-state electrons to higher excited-state energy levels, and the energy level of the intermediate excited state could be used as a bridge for the energy transfer to achieve multi-photon emission. As one of the rare-earth ions in visible light emission, Tb^3+^ has a large energy gap between the ^5^D_4_ excited state and the ground state for less influence on the multi-phonon relaxation [6]. The large proportion of green-light emissions from the ^5^D_4_ to ^7^F_5_ transition makes the ^5^D_4_ level of Tb^3+^ meet the requirement of efficient quantum cutting [1,7]. In efficient quantum cutting of Tb^3+^, the appropriate host environment and intermediate excited state are used to achieve energy level matching between two different or identical ions for energy transfer. In the research of single-ion quantum cutting of Tb^3+^, the quantum efficiencies of K_2_YF_5_:Tb^3+^ [1], Ca_9_Y(PO_4_)_7_:Tb^3+^ [8], and Ba_9_Lu_2_Si_6_O_24_:Tb^3+^ [9] excited at 245, 222, and 251 nm were 121.3%, 157.2%, and 144%, respectively. In the level matching, the ^5^D_4_ energy level of Tb^3+^ was directly used as an intermediate excited state to transfer the excited-state electrons [8]. If both ^5^D_3_ and ^5^D_4_ excited states of Tb^3+^ participate in the ^7^F*_J_* (*J* = 6, 5, 4, 3) luminescence emission at the same time, there is an emission competition between the two excited levels [7], and the emission efficiency of ^5^D_4_ level in the visible region is reduced. On the other hand, energy transfer can also be realized with the aid of other ions for a suitable level matching. In the CeO_2_:Tb^3+^/Yb^3+^ [10], BaGdF_5_:Tb^3+^ [11], and NaGdF_4_:Ho^3+^/Yb^3+^ [2] materials, Yb^3+^and Gd^3+^ could be used as an energy transfer bridge to achieve quantum cutting efficiency of 164, 177, and 179.8%, respectively. To sum up, the direct energy transfer processes, an appropriate host, and energy level matching are necessary key conditions to achieve high efficiency of quantum cutting in luminescence materials.

In Stokes emission, most excitation wavelengths of quantum cutting are located in the short-wave or vacuum ultraviolet regions for visible light emission, such as 142 nm [12], 222 nm [13], and 251 nm [9]. However, there are few reports on quantum cutting excited by middle-wave ultraviolet light due to the limitation of exciting photon energy and the lack of matching energy levels. Middle-wave ultraviolet light has relatively good penetrability in air, resin, or glass. If the quantum cutting phosphor excited by ultraviolet B is developed, the application of phosphor in lamps, decoration, and anti-counterfeiting could be greatly improved and expanded [1]. We studied the luminescent spectra of Tb^3+^ in KTb(CO_3_)_2_ system in detail [14]. The larger excitation intensities exist in the ultraviolet B–excited spectrum, which has a potential application in quantum cutting for efficient luminescence. Compared with Tb, the rare-earth Y has a relatively cheaper price, which has a similar ionic radius and the zero electron of 4f configuration [15,16]. It is possible for Y^3+^ to completely substitute Tb^3+^ in KTb(CO_3_)_2_ as a luminescent host. In the luminescent emission of Tb^3+^-doped KY(CO_3_)_2_, the energy of the exciting photons could be further reduced by the emission transitions of the excited electrons to the lower ^5^D_4_ excited state. In this contribution, quantum cutting, spectral characterizations, and decay time of Tb^3+^-doped KY(CO_3_)_2_ were investigated in short- and middle-wave ultraviolet ranges.

## 2. Materials and Methods

KY_1−x_(CO_3_)_2_:*x*Tb^3+^ (KYC:*x*Tb^3+^) phosphors were synthesized by the hydrothermal method [14]. The raw materials of Tb(NO_3_)_3_·6H_2_O (99.99%) and Y(NO_3_)_3_·6H_2_O (99.99%) were bought from the Shanghai xianding Biotechnology Co., Ltd. (Shanghai, China). First, (1 − x) mmol Y(NO_3_)_3_·6H_2_O and *x* mmol Tb(NO_3_)_3_·6H_2_O were dissolved into 3 mL of deionized water for the mixture solution. Then, the mixture was added dropwise to the 25 mL K_2_CO_3_ solution (0.55 mol/L) under vigorous stirring. The pH value of the mixed solution was adjusted to 9.5 by the dilute nitric acid. The final solution was transferred to the 50 mL Teflon autoclave, which was reacted at 200 °C for 8 h with a heating rate of 5 °C/min. Deionized water and ethanol were used to wash the precipitate. The phosphor was prepared after drying at 60 °C for 40 min in the air.

A quick response code (QR code) was prepared by the epoxy resin and phosphor. First, the phosphor could be dispersed well in epoxy resin by stirring. After the two substances were fully mixed, the mold was used to fabricate the desired object under heating at 100 °C for 60 min in the air.

The crystal structures were analyzed by the X-ray powder diffraction in the range of 10–70° (PANalytical, Almelo, The Netherlands). The morphologies and energy dispersive spectrum (EDS) were imaged with a cold field emission scanning electron microscopy (Regulus 8220, Hitachi High-Tech Co., Tokyo, Japan). The excitation spectra and emission spectra were measured by a FLS920 fluorescence spectrophotometer (Edinburgh Instruments, Livingston, UK) with the excitation of 450 W Xe-lamp. Using BaSO_4_ as a reference, the absolute quantum efficiency (QE) was measured by the integrating sphere within the FLS920 sample chamber in a direct and indirect method. The decay curves were tested with the use of the 60 W microsecond flashlamp (Edinburgh Instruments, Livingston, UK).

## 3. Results and Discussion

### 3.1. Crystal Structures

Figure 1 shows the X-ray diffraction patterns of KYC:*x*Tb^3+^ samples. When *x* increased from 0 to 1, the diffraction peaks of KYC:*x*Tb^3+^ were consistent with that of monoclinic KDy(CO_3_)_2_ (JCPDS:1-88-1423) [17]. There were no foreign diffraction peaks, and it indicated that Tb^3+^-doped KYC is a pure monoclinic phase. In the KYC lattice, Tb^3+^ and Y^3+^ with similar ion radii could be substituted for each other in any proportion. On the right side of Figure 1, when *x* = 0, the value of KYC diffraction angle at (002) was larger than that of KDy(CO_3_)_2_ because the radius of Y^3+^ (1.019 Å) was smaller than Dy^3+^ (1.027 Å) [16]. As *x* increased, the diffraction peak corresponding to the (002) crystal plane was shifted toward a small angle. Therefore, the variation of cell parameters indicated that the environment of Tb^3+^ is modified slightly in the KYC lattice.

### 3.2. Morphologies and Element Analysis

The morphologies of KYC:*x*Tb^3+^ at different *x* values are shown in Figure 2. When *x* value was 0.01, the phosphor exhibited as a single monoclinic particle with a size of 40–60 μm in Figure 2a. With the increase in Tb^3+^ ion concentration, the growth rate of the nucleus was obviously accelerated. Small monoclinic crystal particles aggregated with each other to form a large particle (Figure 2b–d). When the *x* further increased to 0.7 and 1, individual grains grew larger and easily fractured into small pieces (Figure 2e–f). A small area was selected from Figure 2d, and energy dispersive spectroscopy of Figure 2g was taken to demonstrate the existence of K, Y, Tb, and O elements in Figure 2h–l. From the element distribution maps, K, Y, and Tb ions were well-dispersed in KYC:0.5Tb^3+^. The relative elemental composition was close to the original stoichiometric ratio.

### 3.3. Luminescence Spectra

Figure 3 shows the luminescence spectra of KYC:*x*Tb^3+^. From the excitation spectra presented in Figure 3a, the whole excitation spectrum of Tb^3+^ was mainly composed of 4f^8^–4f^7^5d^1^ transition in the range of 200–300 nm and 4f^8^–4f^8^ transition in the range of 300–390 nm [3,18,19]. In the f–d transition, there were mainly two strong excitation peaks at 245 and 283 nm, corresponding to the transition from the ^7^F_6_ ground state to ^7^D*_J_* and ^9^D*_J_* levels [9], respectively. When the Tb^3+^ ion doping concentration is low, the excitation intensity corresponding to ^7^D*_J_* is much greater than ^9^D*_J_* because ^7^D*_J_* transition is spin-allowed and ^9^D*_J_* transition is spin-prohibited [9]. As the doped concentration of Tb^3+^ increases, the relative exciting intensity of ^7^D/^9^D*_J_* is decreased due to the sensitive f-d transition in the variation of KYC:Tb^3+^ crystal field [20]. When the concentration of Tb^3+^ ion was greater than 10%, the ^9^D*_J_* forbidden transition of Tb^3+^ was gradually abolished. The intensity of the excitation peak at 283 nm increased significantly in KYC:Tb^3+^, and it indicated that ultraviolet B could be used for efficient excitation. Compared with ultraviolet C and vacuum ultraviolet, ultraviolet B has higher transmittance in glass, resin, and polymer materials. The longer-wavelength UV light is more suitable for the excitation of luminescence emission of KYC:Tb^3+^ in the luminescent products.

Low doping concentration is convenient to characterize the transition process of a single excited-state electron between different levels. Figure 3b shows the emission spectra of Tb^3+^-doped KYC at *x* = 0.001. From the spectra, the intense emission peaks were mainly located in the range of 480–640 nm over the whole spectrum for the exciting wavelength of 245 and 283 nm, corresponding to the transition of ^5^D_4_–^7^F*_J_*, respectively [21]. In the literature, it can be often found that there is a competitive emission between ^5^D_3_ and ^5^D_4_ of Tb^3+^ at low concentrations during the de-population process of the excited electron state [11]. A larger emission region reduces the proportion of green-light emission intensity [22]. When 283 nm was adopted as the excitation wavelength, only two emission bands were located at 376 and 543 nm. Two different emissions indicate that the electrons in the highest excited state are decayed to the ground state through at least two intermediate excited states. When 351 nm was used as the excitation wavelength, these two kinds of emission bands could also be observed. As it is shown in Appendix A, the broadband emission peak intensity at 376 nm gradually decreased with the increase in Tb^3+^ doping concentration. When 376 nm was directly used as the excitation wavelength, the excited-state electron could relax to ^5^D_4_ level to achieve luminescence emission. A suitable crystal lattice environment is also a key factor for the intense emission [23]. Therefore, in the KYC host, especially for excitation at 245 nm of Tb^3+^, there is no competitive emission between ^5^D_3_ and ^5^D_4_ levels, and the high efficiency of Tb^3+^ in the KYC host has more advantage in green-light emission than that of other hosts, owing to the single and direct energy transfer level of ^5^D_4_.

As is shown in Figure 3c, the emission intensity of the green light at 543 nm was increased rapidly with the increase in Tb^3+^-doped concentration in KYC. When *x* was larger than 0.3, the increase rate of 543 nm emission intensity was slowed down. When we used 351 nm as the excitation wavelength, the luminescence was similar to that of 283 nm excitation, but the emission intensity was slightly lower, which was about 50% of that of 283 nm under the same conditions in Figure 3d. The emission intensity was more intense under the 283 nm excitation than under 351 nm due to the larger excitation intensity in Figure 3a excitation spectra. Meanwhile, we also found that KYC:1Tb, where Tb^3+^ was completely substituted for Y^3+^, could emit strong green light without any concentration quenching [14].

### 3.4. Energy Level Diagram

Figure 4 gives a schematic diagram of the electron transition on the excited-state energy level of Tb^3+^. As can be seen from the diagram, electrons at the ^7^F_6_ level were excited to the ^5^K_8_ level by the 245 nm excitation light. The energy levels ^5^K_8_, ^5^F_4_, ^5^D_4_, ^7^F_0_, and ^7^F_6_ of Tb^3+^ were at 40,749, 35,380, 20,545, 5703, and 74 cm^−1^, respectively [24]. The ^5^K_8_→^5^D_4_ energy gap of 20204 cm^−1^ was similar to that of ^7^F_6_→^5^D_4_ (20471 cm^−1^). The excited-state electrons could transfer energy directly to the ^5^D_4_ level by the cross-relation: ^5^K_8_ + ^7^F_6_→^5^D_4_ + ^5^D_4_ (process 1), and then, the two excited-state electrons transferred back to the ^7^F*_J_* (*J*=6, 5, 4, 3) for the quantum cutting emission at 493, 543, 583, and 623 nm, respectively. Under the excitation at 283 nm, the ground-state electrons of ^7^F_6_ were excited to the ^5^F_4_ level [1]. Then, the excited-state electrons returned to the two lower excited states ^5^G_6_ and ^5^D_4_. A proportion of excited-state electrons could transfer in a way of ^5^F_4_ and ^5^G_6_ to ^7^F*_J_* with a board emission (360–420 nm), which is clearly shown in Figure 3b. It is worthwhile to note that the energy gaps were almost the same between ^5^F_4_→^5^D_4_ (14835 cm^−1^) and ^7^F_0_→^5^D_4_ (14842 cm^−1^). The other proportions of excited-state electrons could more easily return to ^5^D_4_ with the cross-relation: ^5^F_4_ + ^7^F_0_→^5^D_4_ + ^5^D_4_ (process 2). Finally, two excited-state electrons of ^5^D_4_ transferred to ^7^F*_J_* in a quantum cutting emission. Under the excitation of 351 nm, the excited electrons at the ^5^L_9_ level could return to the ^5^G_6_ level through vibration relaxation and then transfer to ^7^F_J_ with the board emission, which is shown in Figure 3b. The other electrons returned to ^5^D_4_ with the cross-relation between ^5^G_6_→^5^D_4_ and ^7^F_6_→^7^F_0_, and the visible light could be emitted for the transition from ^5^D_4_ to ^7^F*_J_*, which was the same as the excitation process of 376 nm.

Table 1 shows the quantum efficiency (QE) of KYC:*x*Tb^3+^ excited at 245, 284, and 351 nm. Since the Xenon lamp has low emission intensity below 250 nm [8], the QE values were measured under the fixed excitation at 245 nm for the Tb^3+^ spin-allowed transition. From the tendency of value variation, the QEs increased rapidly with the increase in Tb^3+^ doping concentration. The higher QE was due to simple and direct transition processes between ^5^G_6_ and ^5^D_4_, which were the most efficient energy transfer processes. When 283 nm was used as an exciting wavelength, the quantum efficiency reached 109% at *x* = 0.3. As the doping concentration continued to increase, the increasing rate of quantum efficiency became slower. The maximum value was 119% for KYC:0.7Tb^3+^. Under the excitation of 351 nm, the QE value could increase with the increase in the Tb^3+^ concentration. From the analysis of QE value, it was concluded that KYC, in comparison with phosphate [8], silicate [9], and fluoride [11], could emit green light with high efficiency by Tb^3+^ doping. In addition, we will further adopt different preparation methods to control the morphology of samples for better luminescence performance in the future [25,26].

### 3.5. Decay Curves

Figure 5a shows the luminescence decay curves at different Tb^3+^ concentrations in KYC:*x*Tb^3+^. As can be seen from the figure, the decay curves for different concentrations of KYC:*x*Tb^3+^ could be single exponential for the emission process. It was indicated that the excitation electrons of ^5^D_4_ state have a longer lifetime than those of the ^5^K_8_ level. As the doping level increased, the cross-relaxation between the two Tb^3+^ became more intense, and the excited-state electrons decayed faster. The variation in lifetime as a function of x is shown in Figure 5b. It is seen that the lifetime of 543 nm green light decreased with the increase in Tb^3+^ content. The variation in lifetime was small when x was less than 0.1. When x was increased, the green emission lifetime at 543 nm decreased rapidly, which indicated that there is a faster energy migration process between the two Tb^3+^ ions in the crystal lattice of KYC. The fast migration process of excited-state electrons enables KYC:Tb^3+^ to be an efficient green-light-emitting phosphor.

## 4. Quick Response Code

Quick response (QR) code is widely used in our life for information identification, health codes, trademark and product introduction, and so on. A highly efficient phosphor could enhance the QR code recognition in poor lighting conditions. Figure 6 shows the application of information identification for the luminescent QR code of KYC:1Tb^3+^ under the 365 nm LED irradiation. The chromaticity coordinate is (0.3387, 0.5916) in Figure 6a. The point position of KYC:1Tb^3+^ phosphor is in green field. Compared to a common QR code (Figure 6b), the luminescent QR code could be more easily identified in the darkness. Moreover, we can use QR codes to store some information or product introduction, especially in places where the space is small but where a lot of information needs to be expressed. The extended information can be easily obtained by scanning a QR code with a mobile phone in Figure 6c, such as “Huaibei Normal University”. In addition, the information can be kept secret by hiding the QR code under the transparent surface of a product. When information needs to be viewed, the QR code is displayed by ultraviolet excitation.

## 5. Conclusions

The KYC:*x*Tb^3+^ (*x* = 0~1) phosphors were prepared by the hydrothermal method. The excited spectra, emission spectra, quantum cutting, and decay curves were measured and discussed. The ultraviolet wavelengths of 245, 283, and 351 nm were used to excite the phosphors. The exciting intensity of 245 and 283 nm (f–d transition) was larger than that of 351 nm (f–f transition). All the emissions were mainly attributed to the transition from ^5^D_4_ to ^7^F_J_ (*J* = 6, 5, 4, 3, 2). Visible quantum cutting of KYC:*x*Tb^3+^ was observed at two excitation wavelengths of 245 nm and 283 nm. The optimal quantum cutting efficiency was ascribed to the simple and direct cross-relaxation transition under the 245 nm excitation: ^5^K_8_ + ^7^F_6_→^5^D_4_ + ^5^D_4_. The longest excitation wavelength of quantum cutting was 283 nm, which had the maximum QE value of 119% in KYC:0.7Tb^3+^. The high-efficiency KYC:Tb^3+^ phosphors could be potentially used in applications related to information identification.

## Figures and Tables

**Figure 1 materials-15-06160-f001:**
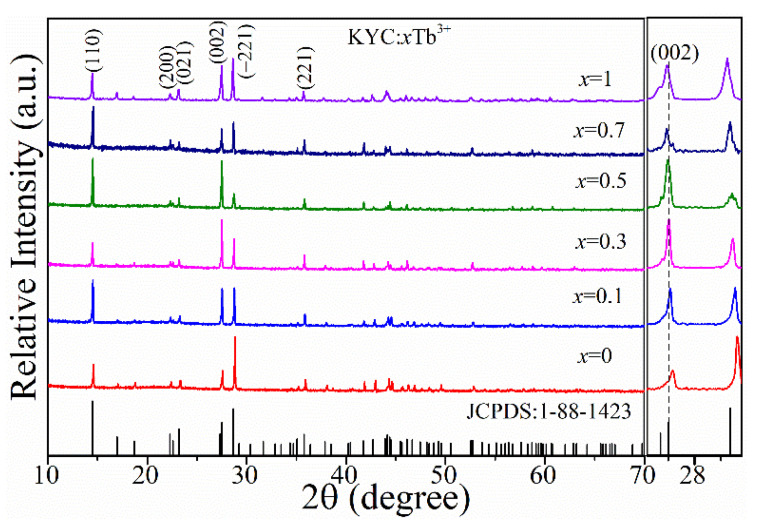
X-ray diffraction spectra of KYC:xTb^3+^ (*x* = 0, 0.1, 0.3, 0.5, 0.7, 1).

**Figure 2 materials-15-06160-f002:**
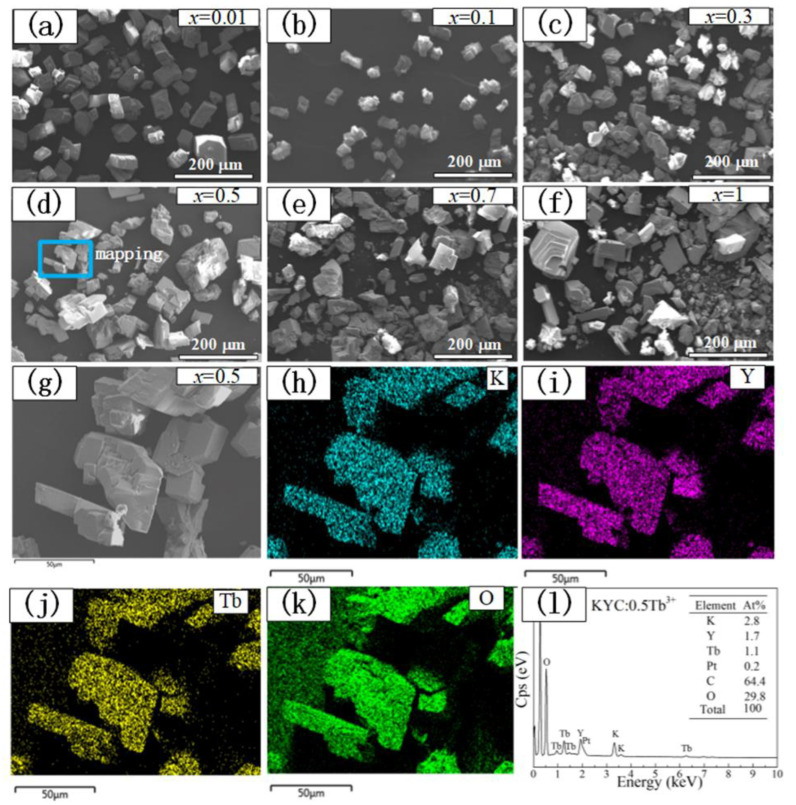
(**a**–**f**) Morphologies of KYC:xTb^3+^(*x* = 0.01, 0.1, 0.3, 0.5, 0.7, 1); (**g**) Selected area of KYC:0.5Tb^3+^; (**h**–**k**) Element mapping of K, Y, Tb, and O; (**l**) Energy dispersive spectroscopy analysis.

**Figure 3 materials-15-06160-f003:**
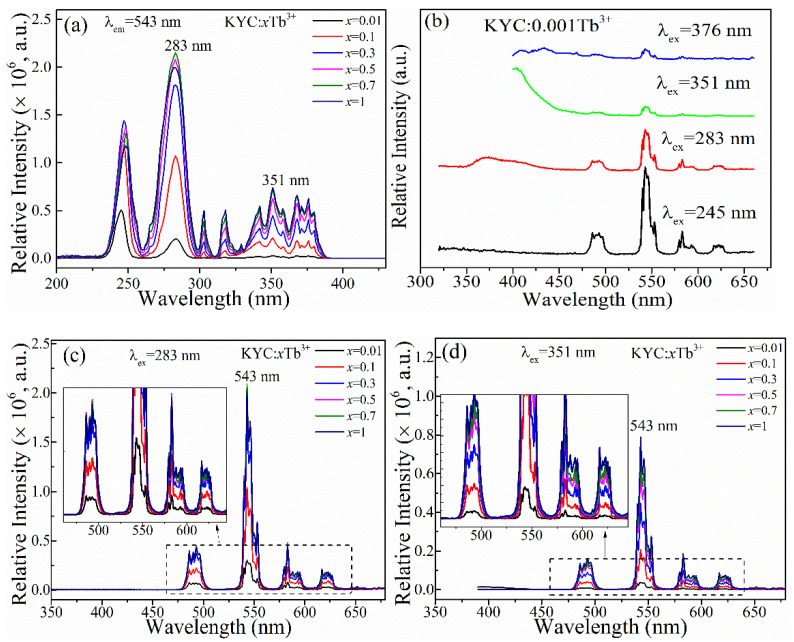
Luminescence spectra of KYC:*x*Tb^3+^: (**a**) excitation spectra; (**b**) emission spectra of KYC:0.001Tb^3+^; (**c**) emission spectra excited at 283 nm; (**d**) emission spectra excited at 351 nm.

**Figure 4 materials-15-06160-f004:**
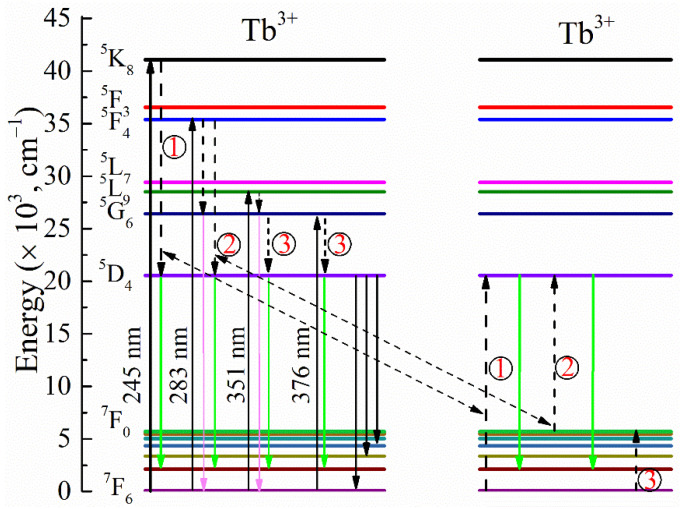
Energy level diagram of KYC:Tb^3+.^

**Figure 5 materials-15-06160-f005:**
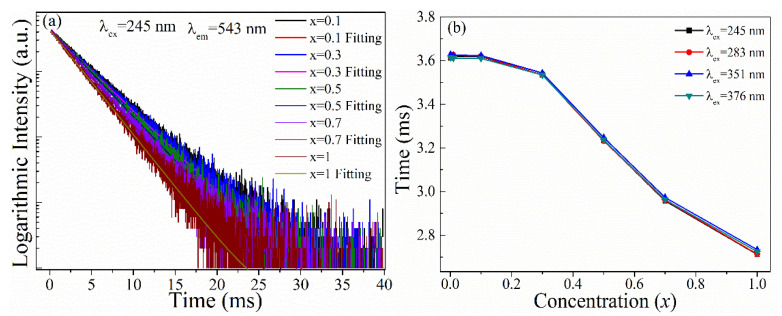
Decay curves (**a**) and energy level lifetimes (**b**) of KYC:*x*Tb^3+.^

**Figure 6 materials-15-06160-f006:**
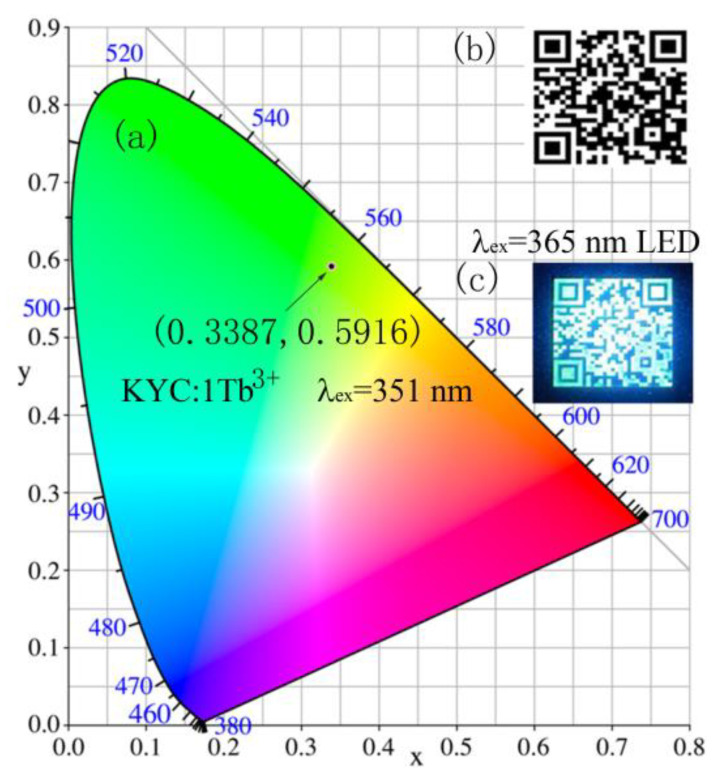
(**a**) Chromaticity point of phosphor; (**b**) Quick response code; (**c**) Quick response code excited by 365 nm LED.

**Table 1 materials-15-06160-t001:** Quantum efficiency of KYC:*x*Tb^3+^ excited at different wavelengths.

Materials	QE245nm	QE283nm	QE351nm	References
*x* = 0.01	138%	23%	5%	This work
*x* = 0.1	172%	70%	22%	This work
*x* = 0.3	177%	109%	43%	This work
*x* = 0.5	171%	115%	64%	This work
*x* = 0.7	160%	119%	67%	This work
*x* = 1	148%	113%	72%	This work
Ca_9_Y(PO_4_)_7_:Tb^3+^	157% (250 nm)			[8]
Ba_9_Lu_2_Si_6_O_24_:Tb^3+^	144% (251 nm)			[9]
K_2_GdF_5_:Tb^3+^	177% (240 nm)			[11]

## Data Availability

The data presented in this study are available on request from the corresponding author.

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
