# Peer review of "Quantum Cutting in Ultraviolet B-Excited KY(CO3)2:Tb3+ Phosphors"

_materials, 2022, doi:10.3390/ma15176160_

Round 1
Reviewer 1 Report
In this article, the authors prepared and tested the optical properties of KY(CO3)2:Tb3+ phosphors. In general, the article can be reconsidered for publication after a major revision.
1) Experimental methodology. Diluted nitric acid usually reduces the pH (below 7), in this regard, it is not clear how the authors achieved pH = 9.5 with the help of nitric acid!
2) Provide morphological analysis, i.e. SEM, TEM images of the samples - how the morphology is changing with the increase of Tb content!
3) Provide EDX elemental mapping, to confirm uniform Tb distribution in a selected area.
4) Supporting information is not available for inspection, looks like the ZIP file is damaged - please attach a pdf or word file instead.
5) Show how the authors measured the quantum efficiency! In this regard, it is better to show the quantum yield of the samples instead!
6) How do CIE coordinates are changing with increasing of Tb concentration?
7) Introduction part can be improved with highly relevant studies such as doi:10.1016/j.matlet.2022.132500 and doi: 10.3390/ma13132987.
Author Response
Dear reviewer:
Thanks for your hard work!
According to the reviewer’s constructive comments, we add the discussions of the morphological analysis and EDS elemental mapping in the manuscript. The corresponding revised contents are marked in the paper.
We appreciate the reviewer’ hard work, and hope the correction could meet with the approval. Once again, thank you very much for your comments and suggestions.
The corresponding point-by-point response, Please see the attachment!

Reviewer 2 Report
This manuscript is interesting and results very satisfying. Few questions was raised in order to improve the quality of this paper.
At Materials and Methods Section, pleases describe how the Quantum Efficiency of samples was determined.
Line 148-151: The comparison between emission spectra of different samples is always a controverse subject, since many physical parameters may interfere in this comparison. However, the point raised by authors can be addressed using the excitation spectra showed in Fig 2(a). Once the excitation spectrum is obtained from the emission of the sample, and if the spectrum is corrected for the emission intensity, than it is clear that the band formed at 283 nm is around twice the intensity of that at 351 nm, meaning the emission at 351 nm is about half of that at 283 nm. I believe this argument is more real than a direct comparison between the emission spectra.
Line 178-179: The quantum efficiency does not depend on the excitation intensity, so this sentence is incorrect.
Author Response
Dear reviewer:
Thanks for your hard work!The constructive comments and suggestions are very important for the improvement of scientific expression in revised manuscript. We have carefully studied the comments, and the main corrections in the paper are as follows:
1、At Materials and Methods Section, pleases describe how the Quantum Efficiency of samples was determined.
Response:
Thanks for your kindly remind.
In our’ paper, we are very sorry for the negligence of the description on the measure instrument of quantum efficiency(FLS920) [The excitation spectra, emission spectra and quantum efficiency were measured by the FLS920 fluorescence spectrophotometer (Edinburgh Instruments) with the excitation of 450 W Xe-87 lamp.]. It is not clear for the QE measure information in the section of Materials and Methods. So, we add the description of integrating sphere in the text.
Revised: “Using BaSO4 as a reference, the absolute quantum efficiency (QE) was measured by the integrating sphere within the FLS920 sample chamber in a direct and indirect method.”
2、Line 148-151: The comparison between emission spectra of different samples is always a controverse subject, since many physical parameters may interfere in this comparison. However, the point raised by authors can be addressed using the excitation spectra showed in Fig 2(a). Once the excitation spectrum is obtained from the emission of the sample, and if the spectrum is corrected for the emission intensity, than it is clear that the band formed at 283 nm is around twice the intensity of that at 351 nm, meaning the emission at 351 nm is about half of that at 283 nm. I believe this argument is more real than a direct comparison between the emission spectra.
Response:
Thanks to the reviewer’ constructive suggestion, the emission intensity is related to several parameters in the measurement process. In the measurement, all tests were performed under the same conditions. The excited spectra can indeed be used to discuss the intensity of excitation.
The revised are as follows: “The emission intensity is more intense under the 283 nm excitation than that of 351 nm due to the larger excitation intensity in Fig. 3(a) excitation spectra.”
3、Line 178-179: The quantum efficiency does not depend on the excitation intensity, so this sentence is incorrect.
Response:
Thank you for your professional guidance in quantum efficiency testing. The light intensity of Xenon lamp is relatively weak below 250 nm, which is a problem that has troubled us for a long time. After receiving the guidance of expert, we could use the integrating sphere to do more research on quantum efficiency in the future.
Original 178-179:“Since Xenon lamp has low emission intensity below 250 nm wavelength [8], the QE values measured at 245 nm excitation wavelength are only for your reference.”
Revised: “Since Xenon lamp has low emission intensity below 250 nm [8], the QE values were measured under the fixed excitation at 245 nm for the Tb3+ spin-allowed transition.”
We appreciate the reviewer’ hard work, and hope the correction could meet with the approval. Once again, thank you very much for your comments and suggestions.

Reviewer 3 Report
My comments are in attachement.

Author Response
Dear reviewer:
The constructive comments and suggestions are very important for the improvement of scientific expression in revised manuscript. We have carefully studied the expression, reference and figure, and these comments are all valuable and helpful for improving our paper and future researches. The main corrections in the paper are as follows:
(1)“The structural and spectroscopic properties were characterized by XRD analysis and fluorescence spectrophotometry, respectively.”
(2) “The results show that the monoclinic crystal structure of KY(CO3)2:Tb3+ remains in the Tb3+ doping range of 0~100%.”
(3) “The Tb3+ content could be selected arbitrarily in KY(CO3)2 host without any concentration quenching.”
(4) “In the luminescent emission of Tb3+ doped KY(CO3)2, the energy of the exciting photons could be further reduced by the emission transitions of the excited electrons to the lower 5D4 excited state.”
(5) “As x increases, the diffraction peak corresponding to the (002) crystal plane is shifted toward small angle.”
(6)“A suitable crystal lattice environment is also a key factor for the intense emission [23].” [Atuchin, V. V.; Aleksandrovsky, A. S.; Chimitova, O. D.; Krylov, A. S.; Molokeev, M. S.; Bazarov, B. G.; Bazarova, J. G.; Xia, Z., Synthesis and spectroscopic properties of multiferroic b’-Tb2(MoO4)3. Opt. Mater. 2014, 36(10), 1631-1635.]
(7) “Meanwhile, we also found that KYC:1Tb, where Tb3+ was completely substituted for Y3+, could emit strong green light without any concentration quenching [14].”
……………….
………………..
We would like to express our great appreciation to you and reviewer for comments on our paper. Once again, thank you very much for your comments and suggestions.
Yours sincerely

Round 2
Reviewer 1 Report
A revised manuscript can be accepted for publication!